# The Influence of Earnings Management and Board Characteristics on Company Efficiency

Hsueh-Li Huang [1], Lien-Wen Liang [2] , Hai-Yen Chang [2,*] and Hsiu-Yuan Hsu [3]

1   Department of Global Business, Chinese Culture University, Taipei 111396, Taiwan;
    shirleyhuang60@gmail.com
2   Department of Banking & Finance, Chinese Culture University, Taipei 111396, Taiwan;
    liang.lienwen@gmail.com
3   Pricewaterhouse Coopers Private Limited, Taipei 111396, Taiwan; tarantp6628@gmail.com
*   Correspondence: irischang1014@gmail.com; Tel.: +886-926293611

**Abstract:** Earnings management is a means by which managers manipulate earnings to conceal the true performance of a company. The characteristics of the board of directors can also influence firm performance. This study applies data envelopment analysis (DEA) and the Tobin regression model to investigate the influence of earnings management and board characteristics on company efficiency. The data sample includes 396 Taiwanese electronics and biotechnology companies from 2009 to 2017. The results indicate that earnings management has an insignificant influence on company efficiency with mixed results on the interactions between earnings management and board characteristics. When companies practiced earnings management, director experiences, a higher proportion of female directors, and a higher number of board meetings increased company efficiency. In contrast, a higher number of independent directors and a higher attendance rate of the directors at the board meeting decreased company efficiency. The results of this study suggest that board diversity, more female directors, and meetings could still improve firm performance despite companies' engagement in earnings management.

**Keywords:** earnings management; board characteristics; window dressing; company efficiency; firm performance

## 1. Introduction

A multitude of fraud cases has occurred in the last decade, which made earnings management attract increasing attention from both regulators and investors [1]. Earnings management refers to the manipulation of corporate profits presented in financial statements. Corporate managers engage in earnings management for many reasons, such as higher bonuses for managers, fewer errors in financial forecasts by analysts, tax savings, provision of positive information to investors, easier access to required capital, and stability of a company's profits and losses over a sustained period. Earnings management tends to cause information asymmetry. If the results presented in the financial statements affect the interests of the company or management authority, then corporate managers would be more inclined to engage in earnings management practices.

Many corporate accounting scandals have been uncovered since 2000. The most notable cases include Enron, a large U.S. energy company, and WorldCom, a giant US. telecommunication company. Enron Corporation was one of the largest U.S. energy companies with hundreds of billions of dollars worth of assets in 2000. Enron Corporation bankrupted within a few weeks in 2002. Similarly, WorldCom, a large telecommunications company in 2002, was involved in one of the most massive accounting frauds in U.S. history, manipulating its stock prices with its financial statements. With the detrimental effect of financial frauds involving Enron and WorldCom, the U.S. enacted a new corporate reform bill in 2002, which was the Sarbanes–Oxley Act (SOA).

Cohen et al. [2] found that after the implementation of SOA, earnings management through accruals tended to arouse investor attention. Earnings management can be commonly divided into four categories. The first category is sales manipulation, which is to manipulate sales volume through sales discount or loose credit requirements to increase short-term sales amount [3]. Roychodhury [4] noted that when companies face pressure to boost revenue, they tend to add short-term sales by offering customers price discounts. The second category is overproduction. To manipulate earnings, managers could produce more units of goods than the expected demand so that the operating cash flow falls below the normal sales level. The third category is the reduction in discretionary expenses and discretionary accruals to achieve short-term sales targets. Hepworth [5] emphasized that corporate management is not to maximize profits but to stabilize net income while avoiding the reporting of falling profits. When corporate managers are stressed to show satisfactory performance, they are more likely to manipulate corporate profits and losses through earnings management techniques. Particularly, when companies are in their maturity stage and experience limited growth, managers tend to engage in earnings management to give their shareholders an incorrect perception that the company can generate higher earnings per share and return on investment and is thus worth the investors' money. The fourth and last category is the manipulation of intangible assets. Intangible assets such as patents and copyrights are critical to electronic and biotechnology companies. Zucca and Campbell [6] found that financially distressed firms practiced one-time write-off of intangible assets during the current period, knowns as the "big bath", to show improvements in the subsequent periods. Similarly, Markarian et al. and Jones [7,8] discovered that firms used intangible assets to achieve earnings management. Specifically, companies capitalized on the costs of intangible assets to reach consistency in earnings over time. Consequently, manipulating the company's financial statements has a considerable impact on the capital markets in which investors select stocks to buy based largely on firms' financial performance.

The Taiwan electronic and biotechnology companies represent the largest sector of the Taiwan economy, accounting for more than 50% of the exports and over 60% of the market capitalization in the Taiwan Stock Exchange since 2000 [9,10]. With rapid growth, Taiwan-listed electronic and biotechnology companies have attracted an increasing number of individual and institutional investors worldwide since 2016 [10]. Therefore, investors must assure that these listed companies use the invested funds properly to achieve high efficiency, which, however, could be distorted by earnings management. Although corporate managers can decide the use of earnings management, these managers are subject to the supervision of the board. Consequently, the board of directors may mitigate the negative effects of earnings management and maintain company efficiency through proper governance.

The accounting fraud cases were mostly caused by inadequate supervision by the board of directors and management's practice of earnings management to serve their own interests. The board of directors evaluates corporate managers' performance by examining operating performance. Consequently, managers with excellent performance are granted higher salaries, commissions, or bonuses. The board of directors provides incentives to managers to align the interests of management and shareholders, but in recent years managers with unusually high salaries and bonuses have been called "fat cat" and aroused tremendous controversy and dissatisfaction among investors. Therefore, governments around the world have attempted to prevent the recurrence of fraud cases and promote corporate governance in business entities.

The board of directors sets a company's mission and business model. Hence, the decisions of the directors and their integrity play an important role in setting the direction of the companies which these directors govern. In recent years, research on board of directors focused on board diversity, including directors' background, characteristics, personalities, and whether the background of the directors affects the effectiveness of the board [11,12].

The extant literature probed into earnings management mainly for the manufacturing and retail industries. This study emphasizes the importance of the largest Taiwan electronic and biotechnology sector with fast-growing market capitalization and an increasing number of international investors. We aim to discover whether proper corporate governance, reflected by the board characteristics, can reduce the negative effect of earnings management of these firms and maintain production efficiency measured by outputs against inputs.

Moreover, prior literature examined the influence of earnings management and the use of financial ratios to indicate firm performance [13]. These authors suggested that poor corporate governance greatly affects earnings management. However, financial ratios are susceptible to the existence of earnings management and thus unable to reflect the true performance of the companies. Despite the large quantity of literature on corporate governance and earnings management, scarce studies examined the relationship between the two factors.

This study fills the gap by investigating the influence the earnings management and the characteristics of the board of directors on firm performance from the perspective of productive efficiency for the main industry in Taiwan. The purpose of this paper is to examine the influence of earnings management and board characteristics of Taiwan-listed electronic and biotechnology companies on production efficiency from 2009 to 2017 using the data envelopment analysis (DEA) [14–16]. Moreover, this study detects the effect of the interaction of earnings management and board characteristics on firm performance.

In addition, this paper uses both financial and non-financial indicators to gauge corporate performance for two reasons. First, financial indicators can be limited by the accounting policies selected by the managers. Moreover, financial indicators can be a result of earnings management and are thus unable to reflect the true conditions of the companies. Second, many corporate decisions are formulated by non-financial factors, such as corporate governance. Therefore, non-financial factors should be considered in the evaluation of firm performance. The results of this study indicate that earnings management has no significant influence on company efficiency. However, when companies practiced earnings management, director experiences, a higher proportion of female directors and a higher number of board meetings increased company efficiency. On the contrary, a higher number of independent directors and a higher attendance rate of the directors at the board meeting decreased company efficiency. The results of this study suggest that board diversity, female directors who attend to details, and frequent board-of-director meetings could improve firm efficiency despite earnings management.

## 2. Literature Review and Hypothesis Development

### 2.1. Earnings Management

Financial statements indicate a company's financial performance. Charoenwong and Ji-raporn [17] stated that earnings play an important role in understanding a company's financial condition and providing essential information for investment decisions. However, corporate managers may manipulate or smooth out earnings on the financial statements through window dressing techniques. Although corporate managers are monitored by the board of directors, investors, clients, and suppliers, the managers are still likely to manipulate corporate earnings to serve their own interests [18,19]. Therefore, detection of earnings management is a critical issue in the field of accounting and finance to uncover the true value of a company.

Francis et al. [20] conversely argued that real earnings management is related to the strength of the country's legal environment. Companies in countries with stricter regulations find it more difficult to engage in earnings management by manipulating discretionary expenses, production costs, and stock buybacks. These authors also discovered that internal ownership and earnings management are negatively correlated. For example, a company with centralized family ownership tends to focus on long-term performance

and thus is unlikely to engage in such costly earnings management that may produce a negative impact on firm performance in the long run.

It could be difficult to capture earnings management directly within a company. When management is motivated to mislead readers of financial statements into believing the financial results are smooth, the management typically selects the accounting method related to the timing of the transaction, discretionary revenue and expense accruals, and non-operating gain and loss to manipulate final profits [21]. Ultimately, corporate managers use their discretionary power to decide discretionary accruals. In particular, managers may use accruals to alter true earnings using a combination of balance sheet and income statement [22]. The accrual adjustments on the income statement alter the current and future earnings [21]. For example, managers commonly use the income statement to practice earnings management [22]. On the other hand, an accrual adjustment can be found in the balance sheet. Such accruals do not affect the company's future earnings and therefore are preferred by most managers [23].

Elyasiani et al. [24] elucidated cases in which business entities invested in other companies. Only investors with high ownership concentration, long investment period, and follow the regulatory agency independent of the investee company are likely to urge the investee companies to reduce earnings manipulations. In addition, institutional investors' heterogeneity in terms of investment duration, ownership concentration, and manager independence are the key factors that need to be considered in the study of institutional ownership effects. Similarly, Lemma et al. [25] found a significantly positive correlation between accrued earnings management and institutional ownership of firms.

### 2.2. Board Characteristics

### 2.2.1. Director Experience

Drobetz et al. [26] analyzed the work experiences of the board of directors. They found that the work experience of external directors is positively correlated with the value of a company. In other words, a board of directors with more years of work experience creates higher firm value.

When the board of directors holds diverse professional expertise and experiences, these directors may complement each other in the group work. Even if the directors possess different views on a specific subject, they could balance each other and improve the effectiveness of the board in the execution of tasks, monitoring of managers, and provision of consultation [27]. Similarly, it is beneficial for the board to include independent directors with different professional backgrounds who may use their networks and resources outside the firm to assist the company's business operations, thereby improving the overall operating performance of the corporation [27,28]. With the inclusion of independent directors, the board may expand its views by allowing more opinions and suggestions related to business operations, because directors who come from different industries may contribute a wide range of management experiences and knowledge [29]. Particularly, a company's board may need directors with accounting, financial, or legal expertise to provide specific opinions to improve the decision-making process. Ultimately, the various views of a diverse board may enhance the company's operating and financial performance [30].

### 2.2.2. Proportion of Female Directors

Lemma et al. [25] found that a gender-balanced board of directors is especially active during the period in which the company's chief executive officer (CEO) turns over. During such a period, increased participation by the board of directors is required to sustain the company's business operation. The findings of Lemma et al. [25] suggested that a gender-balanced board of directors demonstrated higher value when board participation was urgently needed. Nielsen and Huse [31] found that companies with a higher ratio of female directors may help formulate board decisions and supervise company operations effectively, thus improving corporate performance [32].

Previous researchers found that female directors produce a significantly positive effect on corporate performance [33]. Particularly, female directors with higher educational levels improve firm performance more significantly. Likewise, Abbott et al. [34] indicated that companies with female directors' experience fewer practices of earnings management.

### 2.2.3. Proportion of Independent Directors

Jensen and Meckling [35] pioneered research regarding independent directors. These authors suggested that a higher number of independent directors on the board could alleviate the agency problem between management and shareholders, thus improving company performance. Similarly, Liu et al. [36] indicated for companies in China with the government as the primary shareholders that independent directors who effectively supervised these companies and maximized shareholder wealth could lessen part of the inefficiency problem. These authors claimed that independent directors positively influence firm performance in China. Furthermore, Schnatterly and Johnson [37] highlighted that independent directors not only improve corporate governance but also enhance investor preferences. Specifically, institutional investors prefer investing in companies with a higher proportion of independent directors. Klein [38] confirmed that independent directors reduce managerial deception.

Prior literature also revealed results to the contrary. Koerniadi and Tourani-Rad [39] found a negative correlation between the proportion of independent directors and firm performance. A higher number of independent directors does not increase firm value but induces a negative effect on firm value. Linhares et al. [40] claimed that when a company is a family business, independent directors on the board and other supervisory committees could only perform limited functions. Similarly, Meng et al. [41] used regulatory changes to examine the impact of independent directors on companies and found that independent directors have a negative impact on firm performance. In addition, these authors asserted that when directors' duties to conduct supervisory and consulting activities are restricted by higher information-searching costs, then the negative impact of the directors on firm performance turns more pronounced.

### 2.2.4. Number of Board Meetings

Board meetings represent the time and opportunity by which the board of directors exercises its authority. Prior literature indicated that companies with satisfactory operating performance hold a lower number of board-of-director meetings, and the board-of-director meetings are organized at a regular interval. On the contrary, a higher number of board-of-director meetings may be a result of the company's unsatisfactory operating performance. Vafeas [42] studied the relationship between the number of board-of-director meetings and company value and found a negative correlation between the two factors. However, Bissessur and Veenman [43] argued that a higher number of board-of-director meetings indicates active participation of the directors and more time to deal with internal issues, which include earnings management.

### 2.2.5. Average Director Attendance Rate

According to Shen and Chih [44], the directors on the board and other supervisory committees are supposed to be concerned with company matters. The most direct manifestation of such concern is to frequently convene and attend board-of-direct meetings. These authors also found that a higher number of board meetings and attendance rates by the directors positively correlate with firm performance.

### 2.3. Control Variables

Nunes et al. [13] applied the probit regressions and dynamic estimators to examine the relationships between the profitability of small-and-medium-sized service companies and various determinants such as age, size, liquidity and long-term debt. Nunes et al. [45] used a specific set of determinants and found their relationships with profitability for the Por-

tuguese service small and medium-sized enterprises (SME). In particular, Nunes et al. [45] used size (logarithm of total sales), liquidity (ratio of short-term debt to current assets), and long-term debt (long-term debt divided by total assets) as variables. Size was used as a variable because it helps firms achieve economy of scale, diversity products, and raise entry barriers to avert rivalry, thus improving profitability. Liquidity which indicates firms' short-term debt-paying abilities was selected as a variable because it helps firms take advantage of investment opportunities and allows firms to deal with future technological changes. Long-term debt allows firms to finance long-term projects which are expected to be highly profitable [45].

Similarly, Soekarno and Kinanthi [46] adopted the discriminant function analysis to measure the performance of U.S. information and communication technology (ICT) companies using financial information. Based on the above literature, this study used current ratio, debt ratio, size, company age, economic growth rate, and inflation rate as control variables.

## 3. Methodology

DEA is an efficiency evaluation method developed from the concept of technical efficiency. This study applies the slacks-based measure (SBM) in the DEA method to calculate production efficiency [47,48]. Tone [47] introduced two types of models in DEA: radial and non-radial. The Charnes–Cooper–Rhodes (CCR) model is a radial model that deals with proportional changes of inputs or outputs. In contrast, the non-radial SBM efficiency model discards the assumption of proportional changes of inputs or outputs and deals with slacks directly in calculating efficiency levels SBM model can solve the real-world problems better because all inputs or outputs do not necessarily change proportionally. For example, if a company uses labor, materials, and invested capital as inputs, some of the inputs are substitutional and do not change proportionally. If non-radial slacks play an important role in evaluating efficiency, the SBM approach aids managers in decision-making when they utilize efficiency as the only proxy for evaluating the performance of DMUs. The SBM model is structured to meet two conditions. First, the SBM model is characterized by unit invariant which means that the measure should be invariant with respect to the units of data. Secondly, the SBM model should be monotone decreasing in each slack in input excess and output shortfall. Following Tone's [47] proposal, previous researchers applied the SBM model to measure efficiency [49–51]. Therefore, the SBM model is superior to radial models in calculating efficiency values in this study.

### 3.1. Data Source and Sample

The DEA method requires inputs and outputs to evaluate efficiency. The input variables of this study included the number of employees, fixed assets, and research and development expenses. The output variables consisted of net operating income and market value. To not be affected by the 2008 global financial crisis, we collected data from 2009 to 2017 from Taiwan Economic Journal (TEJ), which is the largest Taiwan financial information company with subscribed online database. The data sample included Taiwan-listed companies in the electronics and biotechnology industries from 2009 to 2017. After removing listed companies with incomplete data, we gathered data for 396 companies.

We gathered the data of inputs and outputs from three sources. First, the input of "number of employees" was sourced from the financial statements of each listed Taiwan company available on the Taiwan Stock Exchange website. Second, other inputs (fixed assets, research and development expenditure, net operating income), all outputs (market value, earnings management), independent variables, director experience, female director) and two control variables (company size and company age) were drawn from TEJ. Company size is used as the control variable because the size of the electronic and biotechnology companies varies largely depending on the history and the business scope of these companies. Third, the data for two remaining control variables (economic growth rate and increase in consumer price index) were retrieved from the Advanced Retrieval Econometric

Modelling system (AREMOS) database established by the Taiwan Economic Data Center, a non-profit organization to provide national statistical data.

### 3.2. Model

The efficiency value is between zero and one. If the ordinal least square (OLS) is used to estimate efficiency, then the coefficients would be biased and inconsistent. Therefore, this study used the common Tobit regression model to analyze the influence of earnings management on efficiency. The Tobit regression model is described in Equations (1) and (2).

$$
\begin{aligned}
E_{it} = {} & \alpha_0 + \alpha_1 DA_{it} + \alpha_{2,k} CG_{it,k} + \alpha_3 WC\_R_{it} + \alpha_4 DBT\_R_{it} \\
& + \alpha_5 SCALE_{it} + \alpha_6 AGE_{it} + \alpha_7 EGR_{it} + \alpha_8 CPI_{it} + \varepsilon_{i,t}
\end{aligned}
\tag{1}
$$

$$
\begin{aligned}
E_{it} = {} & \beta_0 + \beta_1 DA_{it} + \beta_{2,k} CG_{it,k} + \beta_3 DA_{i,t} CG_{it,k} + \beta_4 WC\_R_{it} + \beta_5 DBT_{Rit} \\
& + \beta_6 SCALE_{it} + \beta_7 AGE_{it} + \beta_8 EGR_{it} + \beta_9 CPI_{it} + \varepsilon_{i,t}
\end{aligned}
\tag{2}
$$

where $E_{it}$ denotes efficiency. $DA_{it}$ denotes earnings management. $CG_{it,k}$ denotes a vector variable of corporate governance, including $EXP_{it}$ as director experience, $GENDER_{it}$ as the proportion of female directors, $ID_{it}$ as the proportion of independent directors, $MT_{it}$ as the number of board meetings, and $BOARD_{it}$ as the average director attendance rate. The control variables are $WC\_R_{it}$ current ratio, $DBT\_R_{it}$ debt ratio, $SCALE_{it}$ company size, $AGE_{it}$ company age, $EGR_{it}$ economic growth rate, and $CPI_{it}$ growth rate of consumer price index. The variables are explained as follows.

### 3.3. Earnings Management

Dechow et al. [52] estimated discretionary accruals as the proxy variables for earnings management based on the modified Jones model. Therefore, this study used discretionary accruals to measure earnings management. The estimation steps are as follows. First, perform a parameter estimation using Equation (3) during the estimation period:

$$
\frac{TA_{it}}{A_{i,t-1}} = \alpha_1 \left( \frac{1}{A_{i,t-1}} \right) + \alpha_2 \left( \frac{\triangle REV_{i,t}}{A_{i,t-1}} \right) + \alpha_3 \left( \frac{PPE_{i,t}}{A_{i,t-1}} \right) + \varepsilon_{i,t}
\tag{3}
$$

where $TA_{it}$ denotes total accrued items of the company i in year t. $A_{i,t-1}$ denotes the total assets of the company i in year $t - 1$. $REV_{i,t}$ denotes the change in net sales of the company i in year t. $PPE_{i,t}$ denotes total fixed assets of the company i in year t.

According to Healy and Wahlen [53], the total accrual is the difference between the net profit of a company's continuing operations and the cash flow from operating activities, as shown in Equation (4):

$$
\frac{TA_{it}}{A_{i,t-1}} = \frac{(Earnings_{i,t} - CFO_{i,t})}{A_{i,t-1}}
\tag{4}
$$

Earnings $_{i,t}$ denotes the net profit of a firm's continuing operation. $CFO_{i,t}$ denotes cash flow from operating activities.

The company's financial data for the first three years were used to estimate the parameter values of the company using the OLS Equation (3), and then estimated parameter values were used in Equation (4) to obtain the non-discretionary accruals of each year during the sample period.

The accrued items are shown in Equation (5).

$$
NDA_{i,t} = \hat{\alpha}_1 \left( \frac{1}{A_{i,t-1}} \right) + \hat{\alpha}_2 \left( \frac{\triangle REV_{i,t} - \triangle REC_{i,t}}{A_{i,t-1}} \right) + \hat{\alpha}_3 \left( \frac{PPE_{i,t}}{A_{i,t-1}} \right)
\tag{5}
$$

where, $NDA_{i,t}$ denotes the non-discretionary accrual of the company i in year t. $\triangle REV_{i,t}$ denotes the change in accounts receivable of the company i in year t. $\triangle REC_{i,t}$ denotes net receivables in year t less net receivables in year $t - 1$ scaled by total assets at $t - 1$. Finally,

we deducted the non-discretionary accruals (NDA) calculated from Equation (5) from the total accruals (TA/A) to obtain discretionary accruals (DA) and regarded the discretionary accruals as absolute. The value ($| DA |$) was used as a proxy variable to measure the company's earnings management level.

### 3.4. Corporate Governance

3.4.1. Director Experience

Bantel and Jackson [54] recommended the board have a high level of education and academic diversity. Moreover, directors who have held multiple job positions gained a reputation, and acquired expertise, experiences, and networks from multiple companies could help reduce a company's agency cost. These directors could reduce company risks and enhance firm values [27,55,56].

3.4.2. Proportion of Female Directors

Haque and Brown [57] analyzed the impact of a director's gender on the internal control of listed companies and discovered that the proportion of female directors has a significantly positive correlation with internal control systems and information communication. An increase in the proportion of female directors not only improves the internal control system and strengthens corporate information communication, but also reduces internal violation of regulations.

3.4.3. Proportion of Independent Director

Prior literature indicated that the independence of the board of directors improves the effectiveness of corporate governance mechanisms and firm performance. A higher number of independent directors helps align the interest of management and shareholders [58,59]. Moreover, independent directors with financial and accounting expertise can assist the board in supervising managers. Abbot et al. [60] found that companies that established their audit committees with at least one independent director with accounting expertise have a lower probability of misrepresentations in the financial statements. This phenomenon is probably because directors with accounting expertise possess a deeper understanding of financial issues, which can prevent the occurrence of errors [61]. However, some scholars found a negative relationship between the proportion of independent directors and firm performance. Koerniadi and Tourani-Rad [39] claimed that independent directors negatively affect company value. Independent directors have a positive impact on company value only when these directors are a minority.

3.4.4. Number of Board Meetings

The board of directors mainly fulfills the duties of supervising the CEO at board-of-director meetings. Board meetings urge CEOs to act in the best interests of the shareholders. In addition, the board of directors formulates corporate strategies, evaluates CEOs, and resolves other major issues through board meetings. Therefore, the number of board-of-director meetings reflects the attitude of the board of directors, affecting the level of corporate governance implementation and firm performance.

3.4.5. Average Director Attendance Rate

The average attendance rate of the directors at the board meeting is calculated as the number of times each director actually attended the board-of-director meetings divided by the total number of meetings. The average is then calculated from the attendance rate of each director. The average attendance rate indicates the level of involvement of the directors in company matters.

*3.5. Control Variables*

3.5.1. Current Ratio

The current ratio refers to the ratio of current assets against current liabilities. It is an indicator of the short-term solvency of a company. Therefore, this study used the current ratio as a control variable proposed by Biddle [62].

3.5.2. Debt Ratio

The extant literature found that the capital structure of listed companies affects firm performance. Xu and Wang [63] asserted that the debt ratio of listed companies in China negatively correlates with return on total assets and return on net assets. Therefore, this study used the debt ratio as a control variable.

3.5.3. Company Size

Loomis et al. [64] claimed that large companies have greater pressure to maintain satisfactory performance, and thus these firms tend to adopt more discretionary accruals. However, Watts and Zimmerman [65] argued that large companies engage in fewer earnings management practices than smaller firms to maintain a good reputation. Watts and Zimmerman [66] found that huge companies attract more attention from others and may be regarded as monopolizing the market. These companies practice earnings management to reduce earnings to avoid antitrust investigation [67].

3.5.4. Company Age

Companies that have been established for a longer period tend to see more stable operations. Prior researchers found that companies with a long history have lower information asymmetry and benefit from more experienced management [68].

3.5.5. Economic Growth Rate

The economic growth rate affects the expansion of the companies in a specific region. Thus, this study used the domestic economic growth rate as a control variable.

3.5.6. Inflation Rate

The inflation rate refers to the increase in the consumer price index in the current year as compared to the prior year. This study used the inflation rate as a control variable to increase the accuracy of the empirical results. Table 1 presents the variables, definitions, and data sources.

**Table 1.** Variables, Definitions, and Data Source.

|  | **Variable** | **Definition** | **Data Source** |
|---|---|---|---|
| Input | Number of employees | Number of employees | Financial statements of each listed Taiwan company. |
|  | Fixed assets | The natural logarithm of a company's fixed assets. | TEJ |
|  | Research and development expenditure | Total amount of funds spent on a company's research and development projects. | TEJ |
| Output | Net operating income | Gross operating income minus sales returns and discount. | TEJ |
|  | Market value | The 250-day average share price of a company multiplied by number of shares outstanding. | TEJ |

**Table 1.** *Cont.*

| | Variable | Definition | Data Source |
|---|---|---|---|
| Independent variable | Earnings management (DA) | Discretionary accruals, as the proxy variable of actual earnings management | Derived from the regression analysis on data from TEJ. |
| | Director work experience (EXP) | Board of directors' work experiences in professional, government, academic, and business management fields. | TEJ |
| | Proportion of female director (GENDER) | A virtual variable. When a director is female, its value is 1 and otherwise 0. | TEJ |
| | Proportion of independent director (ID) | Calculated as the number of independent directors divided by the total number of directors at the year-end. | TEJ |
| | Number of board meetings (MT) | The natural logarithm of the total number of board meetings during the current year. | TEJ |
| Control Variable | Average meeting attendance rate (BOARD) | Calculated as the actual number of attendance divided by the total number of meetings attended for each director. The average attendance rate of all directors is then computed. | TEJ |
| | Current ratio (WC_R) | Current assets divided by current liabilities. | TEJ |
| | Debt ratio (DBT_R) | Total liabilities divided by total assets. | TEJ |
| | Company size (SCALE) | The natural logarithm of total assets at the current year-end | TEJ |
| | Company age (AGE) | The natural logarithmic of the number of years for which a company has been established. | TEJ |
| | Economic growth rate (EGR) | economic growth rate | AREMOS (Advanced Retrieval Econometric Modeling System) |
| | Increase in Consumer Price Index (CPI) | The annual increase rate in Consumer Price Index | AREMOS |

## 4. Results and Discussion

### 4.1. Descriptive Statistics

We first conducted correlation coefficient analysis to determine whether the problem of collinearity existed due to a high correlation among the selected variables for earnings management, board characteristics, and economic indicators. Table 2 shows the correlation coefficient estimates. The results indicated that the correlation coefficients among all the variables are below 0.6. Therefore, the problem of collinearity does not exist.

**Table 2.** Correlation Coefficient of the Variables.

| | DA | EXP | GENDER | ID | MT | BOARD | WC_R | DBT_R | SCALE | AGE | EGR | CPI |
|---|---|---|---|---|---|---|---|---|---|---|---|---|
| DA | 1 | | | | | | | | | | | |
| EXP | −0.013 | 1 | | | | | | | | | | |
| GENDER | −0.007 | 0.014 | 1 | | | | | | | | | |
| ID | 0.034 | −0.285 | −0.002 | 1 | | | | | | | | |
| MT | 0.016 | −0.019 | −0.016 | −0.038 | 1 | | | | | | | |
| BOARD | 0.012 | −0.053 | −0.025 | 0.221 | 0.202 | 1 | | | | | | |
| WC_R | −0.022 | −0.051 | −0.016 | 0.008 | −0.086 | 0.062 | 1 | | | | | |
| DBT_R | 0.040 | 0.107 | 0.017 | −0.006 | 0.098 | −0.091 | −0.474 | 1 | | | | |
| SCALE | 0.023 | −0.043 | −0.047 | 0.044 | 0.103 | 0.094 | −0.241 | 0.343 | 1 | | | |
| AGE | −0.118 | 0.128 | −0.021 | −0.314 | −0.018 | −0.112 | −0.028 | 0.104 | 0.016 | 1 | | |
| EGR | −0.007 | 0.079 | 0.003 | −0.185 | 0.011 | −0.105 | −0.023 | −0.009 | −0.007 | −0.043 | 1 | |
| CPI | −0.019 | 0.008 | 0.113 | −0.122 | 0.009 | −0.029 | −0.023 | 0.012 | −0.006 | −0.037 | 0.146 | 1 |

The descriptive statistics of the variables in terms of efficiency are shown in Table 3. The efficiency value is between 0 and 1. A value closer to 0 is less efficient, whereas a value closer to 1 is more efficient. The values derived from the SBM model fall between 0.008 and 1.0. The mean of the SBM values is 0.170.

**Table 3.** Descriptive Statistics of Variables.

| Stats | Mean | Variance | Min | Max |
|:---:|:---:|:---:|:---:|:---:|
| SBM Model | 0.170 | 0.190 | 0.008 | 1.000 |
| DA | 0.004 | 0.021 | 0.000 | 1.067 |
| EXP | 0.897 | 0.098 | 0.204 | 1.000 |
| GENDER | 0.169 | 0.174 | 0.000 | 1.000 |
| ID | 0.246 | 0.166 | 0.000 | 0.666 |
| MT | 0.823 | 0.158 | 0.000 | 1.518 |
| BOARD | 0.841 | 0.136 | 0.000 | 1.000 |
| WC_R | 280.647 | 372.248 | 26.730 | 8646.710 |
| DBT_R | 39.595 | 16.884 | 0.900 | 99.760 |
| SCALE | 6.900 | 0.608 | 4.837 | 9.532 |
| AGE | 1.348 | 0.236 | 0.000 | 1.851 |
| EGR | 3.360 | 2.755 | 0.810 | 10.630 |
| CPI | 0.985 | 0.633 | −0.300 | 1.930 |

This study applied the Tobit regression analysis to investigate the influence of earnings management and board characteristics on company efficiency. Particularly, we examined the interactive effect of earnings management and board characteristics on firm efficiency. The results of this study are presented below.

*4.2. Earnings Management (DA)*

Douglas and Wier [69] claimed that corporate managers select accounting practices to maximize their interests or their company's market value, known as earnings management. This study investigated the relationship between earnings management and company efficiency. The empirical results are shown in Table 4.

Based on the results of Column (1), earnings management has an insignificantly positive influence on company efficiency. This outcome means that the use of earnings management has no significant on company efficiency, which indicates firm performance. Such findings contradict the literature that earnings management has a negative influence on firm performance [17–19] probably due to the nature of the Taiwan electronic and biotechnology companies that strive to become global leaders, thus emphasizing business results and performance more than companies in other industries.

We further examined the interaction between earnings management and board characteristics. The results were mixed. With the interaction of earnings management and the three board characteristics of director experience, the proportion of female directors, and the number of board-of-director meetings, earnings management has a significantly positive influence on company efficiency. This outcome suggests that earnings management strengthens firm performance with board diversity, a higher proportion of female directors, and more frequent board-of-direct meetings.

With the interaction of earnings management and the two board characteristics of the proportion of independent directors and average director attendance rate, earnings management has a significantly negative influence on company efficiency. Such a result implies that earnings management can significantly weaken firm performance with more independent directors and a higher attendance rate. Detailed explanations are provided below. Table 4 presents the results of Tobit Regression Analysis using the SBM model.

**Table 4.** Results of Tobit Regression Analysis.

| SBM_Value | (1) | (2) | (3) | (4) | (5) | (6) |
|---|---|---|---|---|---|---|
| DA | 0.135 | −9.271 *** | −1.024 *** | 1.863 ** | −2.788 ** | 14.455 *** |
| | (0.87) | (−2.38) | (−3.21) | (2.04) | (−2.22) | (5.85) |
| EXP | −0.039 | −0.087 ** | −0.037 | −0.037 | −0.038 | −0.036 |
| | (−1.08) | (−2.12) | (−1.04) | (−1.04) | (−1.06) | (−1.02) |
| GENDER | 0.034 * | 0.035 * | −0.006 | 0.035 * | 0.033 * | 0.035 * |
| | (1.80) | (1.81) | (−0.30) | (1.81) | (1.75) | (1.82) |
| ID | 0.034 | 0.032 | 0.033 | 0.050 ** | 0.034 | 0.035 |
| | (1.45) | (1.37) | (1.40) | (2.01) | (1.44) | (1.47) |
| MT | 0.048 ** | 0.047 ** | 0.043 ** | 0.045 ** | 0.026 | 0.037 * |
| | (2.17) | (2.15) | (1.97) | (2.05) | (1.09) | (1.67) |
| BOARD | −0.065 *** | −0.064 *** | −0.064 *** | −0.063 *** | −0.063 *** | 0.003 |
| | (−2.50) | (−2.45) | (−2.45) | (−2.40) | (−2.40) | (0.11) |
| DA × EXP | | 10.691 ** | | | | |
| | | (2.42) | | | | |
| DA × GENDER | | | 9.804 *** | | | |
| | | | (4.16) | | | |
| DA × ID | | | | −4.263 * | | |
| | | | | (−1.92) | | |
| DA × MT | | | | | 3.995 ** | |
| | | | | | (2.34) | |
| DA × BOARD | | | | | | −14.579 *** |
| | | | | | | (−5.81) |
| WC_R | 0.000 *** | 0.000 *** | 0.000 *** | 0.000 *** | 0.000 *** | 0.000 *** |
| | (11.84) | (11.82) | (11.88) | (11.86) | (11.78) | (11.77) |
| DBT_R | −0.001 *** | −0.001 *** | −0.001 *** | −0.001 *** | −0.001 *** | −0.001 *** |
| | (-3.97) | (−4.04) | (−4.11) | (−4.09) | (−4.10) | (−4.48) |
| SCALE | 0.022 *** | 0.022 *** | 0.023 *** | 0.022 *** | 0.022 *** | 0.022 *** |
| | (3.55) | (3.52) | (3.68) | (3.60) | (3.60) | (3.53) |
| AGE | −0.060 *** | −0.060 *** | −0.061 *** | −0.058 *** | −0.058 *** | −0.051 *** |
| | (−3.65) | (−3.66) | (−3.70) | (−3.55) | (−3.52) | (−3.09) |
| EGR | 0.003 ** | 0.003 ** | 0.003 ** | 0.003 ** | 0.003 ** | 0.003 ** |
| | (2.23) | (2.26) | (2.25) | (2.24) | (2.24) | (2.23) |
| CPI | −0.012 ** | 0.013 ** | −0.012 *** | −0.012 ** | −0.012 ** | −0.012 *** |
| | (−2.35) | (−2.43) | (−2.34) | (−2.30) | (−2.31) | (−2.38) |
| cons | 0.137 ** | 0.183 ** | 0.140 ** | 0.126 * | 0.146** | 0.072 |
| | (2.11) | (2.70) | (2.16) | (1.93) | (2.25) | (1.10) |
| sigma_u | 0.067 *** | 0.067 *** | 0.067 *** | 0.067 *** | 0.067 *** | 0.067 *** |
| | (14.46) | (14.33) | (14.46) | (14.33) | (14.32) | (14.39) |
| sigma_e | 0.173 *** | 0.173 *** | 0.173 *** | 0.173 *** | 0.173 *** | 0.172 *** |
| | (70.05) | (70.05) | (70.05) | (70.05) | (70.05) | (70.06) |
| N | 2970 | 2970 | 2970 | 2970 | 2970 | 2970 |

Note: *: Significance level at 10%, **: Significance level at 5%, ***: Significance level at 1%.

### 4.3. Board Characteristics

#### 4.3.1. Director Experience (EXP)

The results in Column (2) in Table 4 show that a director's experience alone has a significantly negative relationship with company efficiency. However, with the inclusion of earnings management, director experience has a significantly positive relationship with company efficiency. This outcome is consistent with Drobetz et al. [26] and Fama [27] that the directors with diverse expertise and experiences may complement one another and reach a consensus on business issues to improve firm performance despite earnings management.

#### 4.3.2. Proportion of Female Director (GENDER)

Based on the results of Column (3) in Table 4, female directors alone have an insignificantly negative relationship with company efficiency. However, with the inclusion of earnings management, female directors have a significantly positive relationship with company efficiency. Such findings correspond to the study by Lemma et al. [25] and Campbell and Minguez-Vera [32] that female directors assist effectively in formulating corporate decisions and improving business operations because female directors are more attentive to details and sensitive to numbers. The female characteristics, therefore, improved the

board's judgments on financial matters. Therefore, a more gender-balance board is more likely to mitigate the negative effect of earnings management.

### 4.3.3. Proportion of Independent Directors (ID)

Based on the results in Column (4) of Table 4, the variable of independent directors by itself has a significantly positive influence on company efficiency. Nevertheless, when earnings management is added, independent directors have a significantly negative relationship with company efficiency. This outcome signifies that when companies practice earnings management, independent directors may not necessarily agree to such an act from their professional perspective, thus reducing firm performance probably due to differences in opinions. This finding is consistent with Klein [38], as independent directors reduce managerial deception, but increase difficulty in reaching a consensus.

### 4.3.4. Number of Board Meetings (MT)

Based on the results of Column (5) in Table 4, when standing alone, the number of board-of-director meetings has a significantly negative relationship with company efficiency. However, with the addition of earnings management, the number of board-of-director meetings has a significantly positive relationship with company efficiency. This phenomenon confirms Bissessur and Veenman's [43] arguments that more board meetings allow timely discussions among the board members and provide up-to-date opinions to resolve internal issues, thus improving firm performance regardless of earnings management.

### 4.3.5. Average Board-of-Director Attendance Rate (BOARD)

Based on the results in Column (6) of Table 4, the average attendance rate of board-of-directors alone has an insignificantly positive relationship with company efficiency. Yet, when earnings management interacts, the variable of average board-of-director attendance rate has a significantly negative relationship with company efficiency. The findings contradict Shen and Chih [44] that a higher meeting attendance rate improved firm performance probably because a greater number of directors at the same meeting express different views, thus unable to reach consensus quickly and neglect the negative effects as a result of earnings management.

### 4.4. Control Variables

This study uses four control variables: current ratio, debt ratio, company size, and company age. The empirical results are described as follows. The current ratio has a significantly positive relationship with company efficiency, which is consistent with McWilliams and Siegel [70] in that higher liquidity of a firm leads to higher firm efficiency. These results agree with Nunes et al.'s [45] findings that liquidity increased profits but only in the firms with high profitability.

On the contrary, the debt ratio has a significantly negative relationship with company efficiency. This outcome indicates that higher debt results in higher liquidity risk, thus lowering potential profitability and firm performance. Such results are consistent with Nunes et al.'s [45] findings that long-term debt is a catalyzing determinant for profitability but only for firms with very low profitability. The results of this are inconsistent with Nunes et al.'s [45] findings that long-term debt restricts other profitable companies from increasing earnings.

Company size has a significantly positive relationship with firm efficiency. This outcome implies that larger companies are better able to achieve economies of scale, thus creating higher efficiency. These results correspond to Nunes et al.'s [45] findings that size is a catalyst of profitability for low-profit firms. The results of this study contradict with Nunes et al. [45] that size becomes completely negligible in highly profitable firms.

However, company age has a significantly negative relationship with firm efficiency. This result indicates that companies that have been established for a longer period tend to respond more slowly to changes and are thus unable to increase efficiency.

This study also used both the domestic economic growth rate and inflation rate as environmental control variables. The two results are described as follows. First, the results of Columns (1) to (6) in Table 4 indicate that the Taiwan economic growth rate has a significantly positive relationship with company efficiency. This outcome shows that a higher economic growth rate in Taiwan increases firm efficiency and thus firm performance. Second, the results of Columns (1) to (6) in Table 4 indicate that the Taiwan inflation rate has a significantly negative relationship with firm efficiency. This result reveals that a higher inflation rate in Taiwan, measured by an increase in the consumer price index of the current year compared to the prior year, reduces firm efficiency.

### 4.5. Robustness Test

This study used the efficiency values derived from the SBM model to conduct the robustness test. The empirical results are shown in Table 5. Based on the results in Table 5, earnings management has an insignificantly positive relationship with firm efficiency.

**Table 5.** Results of Robustness Test.

| SBM_Score | (1) | (2) | (3) | (4) | (5) | (6) | (7) | (8) | (9) | (10) | (11) |
|---|---|---|---|---|---|---|---|---|---|---|---|
| DA | 0.138 (0.89) | 0.136 (0.88) | −9.547 *** (−2.45) | 0.141 (0.91) | −1.054 *** (-3.31) | 0.139 (0.89) | 2.066 ** (2.27) | 0.135 (0.87) | −2.990 (−2.38) | 0.137 (0.89) | 14.805 *** (6.01) |
| EXP | | −0.051 (−1.47) | −0.100 *** (−2.51) | | | | | | | | |
| DA × EXP | | | 11.006 *** (2.49) | | | | | | | | |
| GENDER | | | | 0.034 * (1.83) | −0.007 (−0.33) | | | | | | |
| DA × GENDER | | | | | 10.115 *** (4.29) | | | | | | |
| ID | | | | | | 0.028 (1.25) | 0.046 * (1.93) | | | | |
| DA × ID | | | | | | | −4.756 ** (−2.15) | | | | |
| MT | | | | | | | | 0.033 (1.59) | 0.011 (0.48) | | |
| DA × MT | | | | | | | | | 4.271 ** (2.50) | | |
| BOARD | | | | | | | | | | −0.048 (−1.93) | 0.018 (0.67) |
| DA × BOARD | | | | | | | | | | | −14.934 *** (−5.97) |
| WC_R | 0.000 *** (11.65) | 0.000 *** (11.65) | 0.000 *** (11.63) | 0.000 *** (11.69) | 0.000 *** (11.75) | 0.000 *** (11.62) | 0.000 *** (11.66) | 0.000 *** (11.71) | 0.000 *** (11.66) | 0.000 *** (11.72) | 0.000 *** (11.69) |
| DBT_R | −0.000 *** (−3.80) | −0.000 *** (−3.61) | −0.000 *** (−3.69) | −0.000 *** (−3.84) | −0.000 *** (−3.99) | −0.000 *** (−3.79) | −0.000 *** (−3.94) | −0.000 *** (−3.86) | −0.000 *** (−4.01) | −0.000 *** (−3.96) | −0.001 *** (−4.51) |
| SCALE | 0.021 *** (3.55) | 0.020 *** (3.39) | 0.020 *** (3.3) | 0.022 *** (3.63) | 0.023 *** (3.75) | 0.021 *** (3.43) | 0.021 *** (3.48) | 0.021 *** (3.47) | 0.021 *** (3.54) | 0.023 *** (3.77) | 0.023 *** (3.74) |
| AGE | −0.064 *** (−4.03) | −0.063 *** (−3.9) | −0.063 *** (−3.94) | −0.064 *** (−4.03) | −0.065 *** (−4.06) | −0.060 *** (−3.67) | −0.058 *** (−3.55) | −0.063 *** (−3.98) | −0.061 *** (−3.83) | −0.066 *** (−4.15) | −0.056 *** (−3.54) |
| EGR | 0.002 ** (2.16) | 0.002 ** (2.28) | 0.002 *** (2.32) | 0.002 ** (2.18) | 0.002 ** (2.21) | 0.002 *** (2.36) | 0.002 ** (2.37) | 0.002 ** (2.14) | 0.002 ** (2.14) | 0.002 * (1.95) | 0.002 (1.93) |
| CPI | −0.011 ** (−2.29) | −0.011 ** (−2.30) | 0.012 *** (−2.36) | −0.012 *** (−2.49) | −0.012 *** (−2.47) | −0.010 ** (−2.13) | −0.010 ** (−2.07) | −0.011 ** (−2.30) | −0.011 ** (−2.26) | −0.011 ** (−2.31) | −0.011 *** (−2.35) |
| cons | 0.107 (2.20) | 0.156 (2.64) | 0.202 (3.27) | 0.099 (2.03) | 0.101 (2.06) | 0.097 (1.97) | 0.086 (1.73) | 0.082 (1.61) | 0.094 (1.84) | 0.143 (2.74) | 0.072 (1.36) |
| sigma_u | 0.067 *** (14.37) | 0.067 *** (14.38) | 0.067 *** (14.39) | 0.067 *** (14.38) | 0.067 *** (14.51) | 0.067 *** (14.40) | 0.067 *** (14.40) | 0.066 *** (14.25) | 0.066 *** (14.25) | 0.067 *** (14.42) | 0.067 *** (14.46) |
| sigma_e | 0.173 *** (70.09) | 0.173 *** (70.08) | 0.173 *** (70.08) | 0.173 *** (70.08) | 0.173 *** (70.08) | 0.173 *** (70.07) | 0.173 *** (70.07) | 0.173 *** (70.07) | 0.173 *** (70.07) | 0.173 *** (70.09) | 0.172 *** (70.10) |
| N | 2970 | 2970 | 2970 | 2970 | 2970 | 2970 | 2970 | 2970 | 2970 | 2970 | 2970 |

Note: *: Significance level at 10%, **: Significance level at 5%, ***: Significance level at 1%.

We also conducted the robustness test to investigate (1) the relationship between earnings management and efficiency, and (2) the interactions among the variables related to board characteristics. The results showed that earnings management has an insignificantly positive relationship with company efficiency. Regarding the interactions among earnings management and board characteristics, board experience, the proportion of female directors, and the number of board-of-director meetings significantly increased the efficiency of the companies that practiced earnings management. In contrast, the proportion of independent directors and the average attendance rate significantly decreased the efficiency of the companies that practiced earnings management. These results are consistent with that of Table 4. Therefore, the results of this study are robust. Table 5 present the results of the Robustness Test using the SBM model.

## 5. Conclusions

Earnings management has been a concern among investors because manipulating earnings prevents investors from obtaining the true financial performance of a company

over time. Although a company's financial statements may appear appealing, they could mislead stakeholders in the process of decision-making by offering inaccurate information. In addition, a company's board of directors plays an important role in governing management. When the board of directors functions properly, it may enhance company efficiency and performance, thus reducing possible risk exposure by the investors.

The empirical results of this study indicate that earnings management has a positive influence on company efficiency, but insignificantly. This outcome suggests that earnings management has only a minor effect on firm performance. However, the results varied when the interactions between earnings management and board characteristics were considered. In companies that practiced earnings management, director experience, the proportion of female directors, and the number of board-of-director meetings significantly increased company efficiency. This result suggests that board diversity, a higher number of female directors, and more meetings improved firm performance despite managers' engagement in earnings management. Such outcome is consistent with prior literature that diverse inputs from the board of directors holding different experiences, greater attention to financial details by female directors, and timely discussion on the business matters by holding frequent meetings, thus mitigating the negative effects of earnings management. In practice, the Taiwan electronic and biotechnology companies may select their board of directors with different expertise and professional experiences, especially female directors, to hold board meetings regularly. Such practices are likely to enhance firm efficiency while mitigating the negative effects of earnings management.

On the contrary, in companies that practiced earnings management, a higher proportion of independent directors and a higher director attendance rate at the board-of-director meetings significantly decreased company efficiency. Such findings indicate that more independent directors and active involvement of the board members with different opinions could increase the difficulties of reaching consensus and do not necessarily increase company efficiency. In practice, the Taiwan electronic and biotechnology companies should refrain from selecting more independent directors than the legal requirement with the knowledge that a higher attendance rate of the board of directors may not lead to effective meetings.

In sum, although the Taiwan electronic and biotechnology companies may practice earnings management, which seems detrimental to the investors, these companies can still demonstrate a high level of company efficiency when the board has directors possessing different professional experiences, more female directors, and arrange a frequent meeting to discuss timely issues.

The results of this study provide an easy way for individual and institutional investors to distinguish firm performance by examining the board characteristics which are accessible from the financial statements of the electronic and biomedical companies, which is the backbone of the Taiwan economy. The findings also benefit the government in setting policies to strengthen corporate governance and protect investors from the adverse effect of earnings management.

The limitation of the study lies in the scope of this study. This research includes only the electronic and biotechnology companies in Taiwan. Hence the results of this study may not be generalized to the same industry in other Asian regions. Future studies may investigate real earnings management that has drawn increasing attention among market analysts. Additionally, transactions between a company and its affiliates, known as the related-party transactions, provide a new way for a company to engage in earnings management. The results of the related-party transactions can be further examined. Finally, future research may compare the quantiles of the distribution of firm efficiency based on board composition, stakeholders, and financial ratios [45].



**Author Contributions:** Conceptualization, H.-L.H.; methodology, L.-W.L.; software, H.-Y.H.; validation, H.-Y.C.; formal analysis, H.-L.H.; investigation, L.-W.L.; resources, H.-Y.H.; data curation, L.-W.L.; writing—original draft preparation, H.-Y.H.; writing—review and editing, H.-L.H.; visualization, L.-W.L.; supervision, H.-L.H.; project administration, H.-Y.H. All authors have read and agreed to the published version of the manuscript.

**Funding:** This research received no external funding.

**Conflicts of Interest:** The authors declare no conflict of interest.

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
