# Peer review of "The Influence of Earnings Management and Board Characteristics on Company Efficiency"

_sustainability, doi:10.3390/su132111617_

Round 1

Reviewer 1 Report

The topic developed in the document is very interesting. However, I would like to make a few comments.

- Between line 46 and 63, when the authors talk about the categories of earnings management, they do not refer to the activation of intangibles that generate more revenue (they are revenues that are not charged) and, therefore, more profit and more sharefolders funds.

- On the other hand, in the sample of companies I do not see that the size of the companies is distinguished and in the biotech companies their sharing is very different depending on the size.

I hope the comments have been useful, good luck.

Author Response

The response to reviewer 1 is in the attached file. 

Reviewer 2 Report

General comment:

The paper addresses a very relevant topic assessing the influence of earnings management and board characteristics on company efficiency. Although it is recognized merit to the research developed so far, there are several aspects that require further improvement.

Specific comments:

In order to improve the global quality of the manuscript, the following recommendations are made available:

  1. The title needs to be revised, using influence instead of impacts, due to the nature of the methodological tools in use, namely the DEA and Tobit models.
  2. In the introductory item, the motivation and the importance of the current analysis need to be outlined.
  3. Again in the introductory item as well as in the presentation of the literature review and hypothesis development, the following reference study: https://doi.org/10.1080/02642060802398853; needs to be incorporated in order to identify the established set of determinants of firm performance.
  4. Section 2. needs to be retitled and revisited as 2. Literature Review and Hypothesis Development, in order to derive directly the research hypotheses from the reference studies.
  5. In connection, the discussion of the findings needs to incorporate the previously referred research hypotheses for being able to contrast them with previous findings.
  6. In Table 1, it is suggested to include a final column presenting the variables’ sources.
  7. In section 4.5. Robustness Checks, the option for using in this study the efficiency values derived from the SBM model, need to be supported by previous references studies, using a similar procedure as a useful and valid one, both in economic and statistical terms.
  8. The final section needs to comprise, conclusions, limitations and implications, including theoretical and practical implications, especially at the level of performance management.

Author Response

The response to reviewer 2 is included in the attached file. 

Reviewer 3 Report

The article proposal meets the scientific criteria.

Author Response

Dear reviewer, 

Thank you very much for your wonderful comment!  

Round 2

Reviewer 2 Report

The manuscript was substantially improved. Nevertheless, one of the suggestions was not completely addressed, namely, the one concerning the need for also including and discussing the set of determinant factors of profitability, based on the following previous reference study:

Paulo Maçãs Nunes, Zélia Silva Serrasqueiro & João Leitão (2010) Are there nonlinear relationships between the profitability of Portuguese service SME and its specific determinants?, The Service Industries Journal, 30:8, 1313-1341, DOI: 10.1080/02642060802398853 

Author Response

Dear reviewer, 

Thank you for your comments.  Please see the attachment for the revisions. 

Hai-Yen Chang 
